# Postoperative CRP Levels Can Rule out Anastomotic Leaks in Crohn’s Disease Patients

**DOI:** 10.3390/jpm12010054

**Published:** 2022-01-05

**Authors:** Moran Slavin, Avigayil Goldstein, Barak Raguan, Yaron Rudnicki, Shmuel Avital, Ian White

**Affiliations:** 1Department of Surgery, Meir Medical Center, Kfar Saba 4428164, Israel; barak.raguan@gmail.com (B.R.); yaron217@gmail.com (Y.R.); shmuel.avital@clalit.org.il (S.A.); 2Sackler Faculty of Medicine, Tel Aviv University, Tel Aviv 6997801, Israel; avygialt@mail.tau.ac.il (A.G.); ian.white@clalit.org.il (I.W.); 3Department of Surgery, Beilinson Hospital, Rabin Medical Center, Petach Tikva 4941492, Israel

**Keywords:** postoperative CRP, anastomotic leak, Crohn’s disease, NPV

## Abstract

Background: In colorectal cancer, C-reactive protein (CRP) levels on postoperative days 3–4 have a strong negative predictive value for an anastomotic leak, with threshold values of ~15 on post-operative day (POD) 3 and ~13 on POD 4. In Crohn’s disease, CRP levels are perceived as unreliable in the postoperative period because of the underlying inflammatory process. The aim of this study was to determine whether postoperative CRP levels can be used to rule out anastomotic leaks in patients with Crohn’s disease and to set CRP threshold values for this population. Methods: This was a retrospective study of a population of Crohn’s disease patients who underwent surgery with bowel anastomoses at a single high-volume center between 1/2012 and 12/2017. The operations were performed by a single colorectal consultant who is an inflammatory bowel disease specialist. Results: Ninety-two operations were performed. A CRP level of 19.56 mg/dL on postoperative day 3 had an area under the curve of 0.865 (sensitivity 88%, specificity 73%) and a negative predictive value (NPV) of 98% for an anastomotic leak. Patients with an anastomotic leak showed a trend towards decreased postoperative albumin levels (p = 0.06). Conclusions: Mean CRP levels and CRP threshold values were indeed higher in the study population compared with those in colorectal cancer patients. Threshold values were set at 20.3 mg/dL on POD 3, 19.5 mg/dL on POD 4 and 16.7 mg/dL on POD 5. These values had high NPVs and can be used to rule out anastomotic leaks in patients with Crohn’s disease after surgery with bowel anastomosis.

## 1. Introduction 

Anastomotic leak (AL) is one of the most feared complications of gastrointestinal (GI) surgery. With rates ranging between 2% and 15%, AL is associated with high morbidity and mortality [1,2,3,4,5,6,7]. There is no specific method to prevent or to predict an AL. In addition, its diagnosis is not always trivial [8,9,10,11]. Abnormal clinical findings or objective physiological parameters may be absent in the early days after surgery [12,13]; a normal CT scan does not eliminate the possibility of intra-abdominal complications, with false-negative rates of ~20% [14], and output from pelvic drains may be unreliable.

In recent years, C-reactive protein (CRP) has been widely studied as an early predictor of septic complications, including AL, after elective colorectal cancer (CRC) surgery [8,9,10,11]. CRP is an acute-phase reactant protein synthesized in the liver. It is a main component in the inflammatory cascade, with a half-life of 19h, which makes it a very sensitive marker of inflammation [4]. It is commonly used as one of the factors influencing the decision of whether or not a patient is suitable for early discharge after surgery, mainly because of its high negative predictive value (NPV) for ALs [15]. 

The 10-year risk of surgery in patients with CD is as high as 50% [16]. CRP levels are routinely used as a marker of disease activity in these patients [17,18]. However, patients with CD are usually excluded from studies on postoperative CRP levels because of their altered inflammatory response and common use of anti-inflammatory medications [19].

The objective of this study was to investigate postoperative CRP levels in patients with CD who underwent surgery with bowel anastomoses and to assess its use in the early diagnosis of ALs. 

## 2. Materials and Methods 

This was a retrospective study of patients with CD who underwent elective, semi-elective and urgent abdominal surgery with bowel anastomosis at Meir Medical Center between 1/2012 and 12/2017. The operations were performed by a single colorectal consultant who is an inflammatory bowel disease (IBD) specialist. Laparoscopic and open bowel resections and “ostomy" reversal surgeries were included in the study. Patients under the age of 18, surgeries with bowel resections without anastomosis and diversion surgeries were excluded.

Patients’ demographics (age, sex and BMI), CD characteristics (anatomic location of the inflamed bowel, past and current medical treatments, past surgeries and extraintestinal manifestations), operative details (indication, laparoscopic, open or converted and site of anastomosis), complications (return to theater and readmission) and length of stay were recorded. All patients received a single dose of prophylactic broad-spectrum antibiotics and pre-/postoperative low-molecular-weight heparin. 

Medical records of all patients were reviewed to obtain the following parameters for the postoperative period: white blood cell count (WBC), platelet count (PLT), albumin level and CRP level. 

### 2.1. Definitions

AL was defined as a defect seen in the anastomosis at reoperation, the presence of feculent fluid in a pelvic drain at the bedside or evidence of free air, fluid or extraluminal contrast around the anastomosis on CT. Other septic complications included the following: pneumonia, urinary tract infection, superficial wound sepsis and line sepsis. Noninfectious complications included myocardial infarction, deep vein thrombosis or pulmonary embolism.

Urgent surgeries were defined as those that took place less than 24 h after nonelective admissions. The indications for these operations were free perforation and intra-abdominal abscess (IAA) not amenable to percutaneous drainage. 

Semi-elective surgeries were defined as operations indicated by CD exacerbations that did not require urgent surgical intervention: small (<5 cm) IAAs, IAAs amenable to percutaneous drainage and ongoing inflammation leading to prolonged use of steroids or bowel obstructions. Patients with IAAs were treated preoperatively with intravenous (IV) or oral (PO) antibiotics for a period of at least two weeks prior to surgery and percutaneous drainage when necessary. Patients with inflammation-induced bowel obstructions were treated with antibiotics and/or systemic steroids for a similar period of time. These patients were ideally operated on two weeks after completing the tapering down of systemic steroid treatment.

Elective surgeries were operations indicated by a stenotic bowel obstruction, a planned “ostomy” reversal in a patient in disease remission or bowel resection due to suspected or proven malignancy. These patients were not under systemic steroid treatment at the time of the operation. 

All semi-elective and elective patients received preoperative nutritional preparation, either orally or parenterally, for 2–3 weeks.

### 2.2. Statistical Analysis

The statistical software package SPSS 20 (IBM) was used to perform statistical analysis. Normality of data was tested by Shapiro-Wilks. The median was used as a measure of the central tendency for continuous variables. Continuous data were assessed using Student’s *t*-test, and the Mann–Whitney *U* test was used for nonparametric data. Pearson’s chi-square test was employed for comparison of categorical variables. A *p* value of <0.05 (two-tailed) was deemed statistically significant.

Receiver operating characteristic (ROC) curve analysis was performed to assess the accuracy of CRP in detecting AL on successive postoperative days. This method involves plotting a curve of sensitivity (true positives) against 1-specificity (true negatives). The accuracy of the test is calculated by measuring the AUC, and the curve itself can be used to identify an optimum cutoff value, which will provide the highest sensitivity and specificity combination to best diagnose the outcome measure. Positive predictive value (PPV) and negative predictive value (NPV) were calculated at the optimum threshold CRP for each day after surgery.

### 2.3. Compliance with Ethical Standards 

This study was approved by the ethics committee of Meir Medical Center. 

All procedures performed in studies involving human participants were in accordance with the ethical standards of the institutional committee and with the 1964 Helsinki Declaration and its later amendments or comparable ethical standards. 

Informed consent was waived by the institutional research committee.

## 3. Results

### 3.1. Patients 

Patients’ demographic data and preoperative inflammatory markers are shown in Table 1. There were no differences in terms of age, gender, BMI, preoperative inflammatory markers or metabolic state between patients who suffered an AL and those who did not. 

### 3.2. Operative and Preoperative Treatment

Fifty-two procedures were laparoscopic, and forty were open or laparoscopic converted to open. Table 2 describes the different types of operations that were performed.

There was no difference in leak rates between the laparoscopic and open groups (4 vs. 7, *p* = 0.15). There were 11 (11.9%) ALs, of which 8 were small bowel to large bowel anastomosis, and 3 were small bowel to small bowel anastomosis (*p* = 0.79). Antibiotic treatment sufficed as the only intervention in three cases, percutaneous drainage was added in three more cases, and five cases required reoperation. Of the five cases that required reoperation, two had a pin-point leak that was treated with the insertion of a T-drain, one underwent resection of the anastomosis and immediate re-anastomosis, and two required take down of the anastomosis and formation of an ileostomy. The mean postoperative day for the diagnosis of an anastomotic leak was 5.3 ± 3.2. Four patients had an EL that was negative for AL: two on POD 4, one on POD 3 and one on POD 5. These patients had other intra-abdominal septic complications: three had an infected hematoma, and one had a minimal amount of pelvic supportive fluid. 

Six operations were urgent (~6%) due to bowel perforation or intra-abdominal abscess not amenable to percutaneous drainage, forty-three operations (47%) were semi-elective (thirty-three due to fistula or intra-abdominal abscesses and ten due to inflammation-induced, recurrent or steroid-dependent bowel obstruction), and forty-three (47%) operations were elective. Only one AL occurred in the urgent operation group, five occurred in the semi-elective group, and five occurred in the elective operation group (*p* = 0.9).

At the time of the operation, 29 patients were treated with systemic steroids, 23 patients were treated with biologic agents (infliximab, adalimumab or vedolizumab), and five patients were treated with a combination of systemic steroids and biologic agents. Other medical treatments included azathioprine, mercaptopurine and 5-ASA derivatives. None of the patients treated with systemic steroids at the time of the operation suffered an AL. 

There were no procedure-related deaths in the study group.

### 3.3. Analysis of Postoperative CRP Levels

Postoperative CRP levels were higher in the AL group on POD 1 (17.9 ± 11 vs. 9.5 ± 5.4, *p* = 0.09) and POD 2 (22.2 ± 10.2 vs.15.7 ± 9.7, *p* = 0.11); however, this difference became significant only on POD 3–5 (see Table 3). Figure 1 demonstrates the difference in CRP levels and trends between the two groups from POD 1 to 5.

ROC curves were produced for POD 3–5 and analyzed to calculate the area under the curve and optimum CRP threshold (see Figure 2). ROC curve analysis revealed POD 3 to be the most predictive of AL, with an AUC of 0.863 for a CRP threshold value of 19.56 mg/dL (sensitivity 88%, specificity 73%). On POD 4, the AUC was 0.805 for a CRP threshold value of 20 mg/dL (sensitivity 66%, specificity 90%). 

The threshold value of 19.56 mg/dL on POD 3 was strongly exclusive of an AL, with an NPV of 98%. A threshold value of 12.5 mg/dL on POD 5 had an NPV of 100%. The PPV trended upward from day to day, reaching 100% on POD 5 for a threshold value of 12.5 mg/dL.

WBC and PLT levels on POD 3 did not show a statistically significant difference between the AL and no-AL groups. The albumin level on POD 3 was lower in the AL group with borderline significance (see Table 4).

## 4. Discussion

While the measurement of postoperative CRP levels has become a standard practice in CRC surgery in many units, this is not the case for patients with CD undergoing surgery with bowel anastomoses. The main reason for this is the premise that these patients have an altered inflammatory response [20] that affects postoperative CRP levels and their interpretation. In addition, patients with CD have elevated baseline CRP levels [17,18], which may also influence postoperative values. Therefore, postoperative CRP threshold values of CRC patients cannot be applied in CD. This study also showed that although postoperative CRP threshold values are higher in the CD population, they can still be safely used as a tool to rule out anastomotic leaks. 

Previous studies have shown that the clinical significance of postoperative CRP measurement is in its NPV for ALs, rather than its PPV [3,7,8,9,10,11,15]. Indeed, this was the case in this study too. The NPV of a CRP level of 19.56 mg/dL on POD 3 was 98%, but the PPV was only 35%. The PPV does increase later in the postoperative period, but by then, other signs of clinical derangement are usually apparent. 

A recent meta-analysis by Yeung et al. summarized the results of 23 studies that assessed the use of postoperative CRP levels as a tool to predict ALs in colorectal surgery [21]. In a day-by-day comparison of the AL groups and the no-AL groups, CRP levels in Yeung’s study were lower than in this one, a trend that was consistent from POD 1 to 5 (see Table 5). 

In addition, threshold CRP values were lower than those reported here: 14.8 mg/dL vs. 19.56 mg/dL on POD 3, 12.3 mg/dL vs. 20.0 mg/dL on POD 4 and 11.5 mg/dL vs. 12.5 mg/dL on POD 5. This finding of relatively elevated postoperative CRP levels in patients with CD correlates with previous reports by Carvello [19] and de Buck [20]. A summary of the differences in CRP values between patients with CD and other colorectal surgery patients is displayed in Table 5.

Low albumin levels in the postoperative period have recently been shown to have a correlation with postoperative complications [22,23]. In this study, postoperative albumin levels showed only a trend toward lower values in patients with AL (3.03 ± 0.5 vs. 2.71 ± 0.4, *p* = 0.06).

This dedicated study is one of the first to address the use of postoperative CRP levels as a tool to rule out ALs in the CD population. Its clinical contribution is in showing that this practice can be implemented not only in the elective surgery setting but also in a heterogeneous group of CD patients that includes emergent cases. This is significant because, in CD, more often than not, patients reach surgical intervention during or soon after disease exacerbation. The limitation of this study is its small cohort size and retrospective nature, which subjects it to selection bias and record-keeping issues.

In conclusion, mean postoperative CRP levels and threshold CRP values are higher in patients with CD undergoing bowel anastomoses compared with patients undergoing operations for CRC. Nonetheless, postoperative CRP levels can be used to rule out ALs in patients with CD. We suggest a threshold of 20.3 mg/dL on POD 3, 19.5 mg/dL on POD 4 and 16.7 mg/dL on POD 5. More dedicated studies on the CD population are required to validate these results.

## Figures and Tables

**Figure 1 jpm-12-00054-f001:**
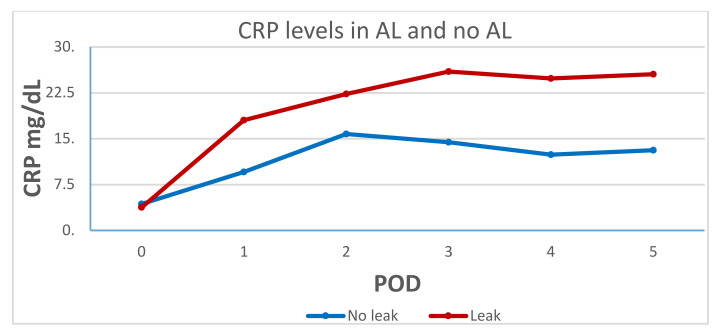
C-reactive protein (CRP) levels in the anastomotic leak (AL) and the no-AL groups.

**Figure 2 jpm-12-00054-f002:**
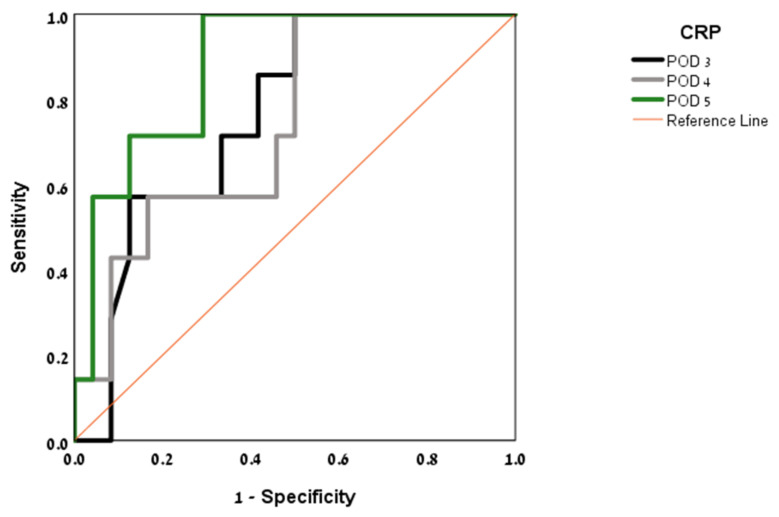
Area under the curve (AUC) by post-operative day (POD).

**Table 1 jpm-12-00054-t001:** Demographics and preoperative inflammatory markers.

	All Patients (*n* = 92)	No Leak (*n* = 81)	Leak (*n* = 11)	*p* Value
Age (years)	40.6 ± 15.7	40.3 ± 15.7	43.3 ± 16.1	0.55
Gender (male/female)	46/46	39/42	7/4	0.52
BMI	22.3 ± 5.5	22.4 ± 5.5	21.9 ± 5.4	0.75
Preoperative WBC (10^3^ cells/µL)	9.0 ± 4	9.0 ± 4.1	8.8 ± 3.7	0.84
Preoperative PLT (10^3^ platelets/µL)	353.4 ± 119.7	351.9 ± 121.8	364.4 ± 107.5	0.74
Preoperative albumin (g/dL)	3.51±0.58	3.51 ± 0.56	3.51 ± 0.82	0.99
Preoperative CRP (mg/dL)	4.68 ± 7.04	4.75 ± 7.32	4.28 ± 5.28	0.84

(BMI: body mass index, WBC: white blood cells, PLT: platelets, CRP: C-reactive protein).

**Table 2 jpm-12-00054-t002:** Types of operations.

Operation	*n* (%)
Right colectomy	33 (35.8%)
Ileocecectomy	27 (29.3%)
Stoma reversal	15 (16.3%)
Small bowel resection	14 (15.2%)
Left colectomy	2 (2.2%)
Subtotal colectomy	1 (1%)

**Table 3 jpm-12-00054-t003:** Postoperative CRP.

	POD 3	POD 4	POD 5
Mean CRP leak	25.9 ± 5.4	24.8 ± 9.6	25.5 ± 10.9
Mean CRP no leak	14.4 ± 8.0	12.3 ± 8.3	13.0 ± 11.1
*p*	0.000	0.001	0.004
AUC	0.863	0.805	0.789
CRP cutoff	19.56	20.0	12.5
Sensitivity	0.88	0.66	1.0
Specificity	0.73	0.90	0.62
PPV (%)	35%	60%	100%
NPV (%)	98%	88%	100%

(CRP: C-reactive protein, AUC: area under the curve, PPV: positive predictive value, NPV: negative predictive value).

**Table 4 jpm-12-00054-t004:** Other inflammatory markers on POD 3.

	No Leak (*n* = 81)	Leak (*n* = 11)	*p* Value
WBC (103 cells/µL)	9.0 ± 3.6	11.0 ± 6.7	0.46
PLT (103 platelets/µL)	319.7 ± 113.1	306.3 ± 116.4	0.59
Albumin (g/dL)	3.03 ± 0.5	2.71 ± 0.4	0.06

(WBC: white blood cells, PLT: platelets).

**Table 5 jpm-12-00054-t005:** Mean CRP levels and CRP threshold values.

	Yeung et al. [21] (Colorectal Surgery)	This Study (CD)	Carvello et al. [19] (CD)
	Leak/No Leak	Leak/No Leak	-
Mean CRP POD 1	11.4 ± 3.25 / 9.58 ± 2.9	17.9 ± 11 / 9.5 ± 5.4	-
Mean CRP POD 2	20.1 ± 2.9 / 14.5 ± 3.1	22.2 ± 10.2 / 15.7 ±9.7	-
Mean CRP POD 3	22.4 ± 5.1 / 12.3 ± 3.2	25.9 ± 5.4 / 14.4 ± 8.0	-
Mean CRP POD 4	20.38 ± 3.8 / 10.5 ± 1.7	24.8 ± 9.6 / 12.3 ± 8.3	-
Mean CRP POD 5	18.7 ± 3.5 / 6.5 ± 2.37	25.5 ± 10.9 / 13.0 ±11.1	-
CRP threshold POD 3	14.8	19.56	21.0
CRP threshold POD 4	12.3	20.0	19.0
CRP threshold POD 5	11.5	12.5	21.0

## Data Availability

The data analyzed in this study are available upon request under instructions of the Institutional Review Board.

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
