# Peer review of "Postoperative CRP Levels Can Rule out Anastomotic Leaks in Crohn’s Disease Patients"

_jpm, 2022, doi:10.3390/jpm12010054_

Round 1
Reviewer 1 Report
The current study addressed an important clinical question: how to predict AL complications after CD surgery using inflammatory cytokines? However, this topic has been extensively studied in previous literatures (Updates Surg. 2020;72(4):985-989.; Zhonghua Wai Ke Za Zhi. 2016;54(8):620-623. ; Inflamm Bowel Dis. 2018;24(9):1992-2000.. Gastroenterol Res Pract. 2021;2021:6629608.) Compared with these studies, the current study did not provide much new information on literatures. Also, I have two comments: 1) Did any patients have postoperative IASC after surgery? For these patients, an elevated CRP was also observed. 2) The demographics of the patients and preoperative parameters (such as preoperative drainage, use of drugs, disease behavior) were not introduced.
Author Response
Dear Sir/Madam,
thank you for your comments.
in regard to comment no.1-
please see an addition about patients with other IASC on page 6 paragraph 3.
in regard to comment no.2-
please see an addition about patients' perioperative parameters on page 6 paragraph 4.
patients' medical treatment are described on page 6 paragraph 5.
sincerely,
the authors
Reviewer 2 Report
The manuscript entitled postoperative CRP levels can rule out anastomotic leaks in Crohn’s disease patients by Moran Slavin et al. describes threshold values of CRP (20.3 mg/dl in POD3, 19.5mg/dl n POD4 and 16.7 mg/dl in POD5) had high NPV’s and can be used to rule out anastomotic leaks in patients with Crohn’s disease after surgery.
This manuscript is interesting. The following points would benefit from further attention.
- Is it clinically meaningful to be able to distinguish the anastomotic leak from CRP levels on postoperative day 3, by which time symptoms (abdominal pain, fever) may have appeared and emergency surgery or other procedures may be necessary?
- What procedures were performed on the 11 patients with anastomotic leak?
- How does the CRP level change after treatment for anastomotic leak?
Author Response
Dear reviewer,
thank you for taking the time to thoroughly review our work. we appreciate the opportunity to improve it with your comments.
in regard to comment no.1-
this is indeed a main issue in postoperative management. the anastomotic leak usually occurs early, however signs and symptoms are smoldering in their presentation and rarely appear as full blown sepsis. they typically appear late, are not always evident on CT and sometimes diagnosed even after the patient has been discharged (please see page 3 paragraphs 1 and 2)
the difference between an early diagnosis of an anastomotic leak and a late one determines the morbidity the patient will suffer and also affects mortality. this is an established principle and therefore the efforts of searching for biologic markers to assist in early diagnosis of this dreaded complication.
in regard to comment no. 2-
please see additions made to page 6 paragraph 3.
in regard to comment no. 3-
CRP is definitely one of the markers we follow when assessing the recovery of patients after treatment for AL. it usually declines rapidly with the proper treatment, however it is a subject for additional focused research.
again, we would like to thank you for your time.
yours sincerely,
the authors
Round 2
Reviewer 1 Report
The current study has proven that postoperative CRP could predict AL after CD surgery, and the results confirmed the findings in previous literatures.
Reviewer 2 Report
The authors respond to all the requests from the reviewer.